# Tribological Behavior and Mechanism of Silane-Bridged h-BN/MoS2 Hybrid Filling Epoxy Solid Lubricant Coatings

**DOI:** 10.3390/nano15050401

**Published:** 2025-03-06

**Authors:** Xiaoxiao Peng, Haiyan Jing, Lan Yu, Zongdeng Wu, Can Su, Ziyu Ji, Junjie Shu, Hua Tang, Mingzhu Xia, Xifeng Xia, Wu Lei, Qingli Hao

**Affiliations:** 1School of Chemistry and Chemical Engineering, Nanjing University of Science and Technology, Nanjing 210094, China; xxp0812@njust.edu.cn (X.P.); jhy24@njust.edu.cn (H.J.); zongdengwu@njust.edu.cn (Z.W.); sucan925@njust.edu.cn (C.S.); jiziyuyx@njust.edu.cn (Z.J.); shu_junjie@njust.edu.cn (J.S.); xiamzh196808@njust.edu.cn (M.X.); 2AECC Guizhou Honglin Aero-Engine Control Technology Corporation Ltd., Guiyang 550009, China; 19985116218@163.com (L.Y.); 18914852207@163.com (H.T.)

**Keywords:** epoxy resin, tribological property, wear, h-BN/MoS_2_, surface modification

## Abstract

To significantly improve the tribological performance of epoxy resin (EP), a novel h-BN/MoS_2_ composite was successfully synthesized using spherical MoS_2_ particles with lamellar self-assembly generated through the calcination method, followed by utilizing the “bridging effect” of a silane coupling agent to achieve a uniform and vertically oriented decoration of hexagonal boron nitride (h-BN) nanosheets on the MoS_2_ surface. The chemical composition and microstructure of the h-BN/MoS_2_ composite were systematically investigated. Furthermore, the enhancement effect of composites with various contents on the frictional properties of epoxy coatings was studied, and the mechanism was elucidated. The results demonstrate that the uniform decoration of h-BN enhances the chemical stability of MoS_2_ in friction tests, and the MoS_2_ prevents oxidation and maintains its self-lubricating properties. Consequently, due to the protective effect of h-BN and the synergistic interaction between h-BN and MoS_2_, the 5 wt % h-BN/MoS_2_ composite exhibited the best friction and wear resistance when incorporated into EP. Compared to pure EP coatings, its average friction coefficient and specific wear rate (0.026 and 1.5 × 10^−6^ mm^3^ N^−1^ m^−1^, respectively) were significantly reduced. Specifically, the average friction coefficient decreased by 88% and the specific wear rate decreased by 99%, highlighting the superior performance of the h-BN/MoS_2_-enhanced epoxy composite coating.

## 1. Introduction

Friction and wear are primary contributors to mechanical equipment failure, leading to material loss and heat dissipation, which result in substantial energy wastage and potentially impede industrial progress [1]. Therefore, mitigating friction and wear by enhancing lubrication performance is a crucial measure for conserving energy, reducing material loss, and maintaining equipment. Compared to traditional lubricants, polymer composite coatings not only exhibit remarkable performance in ambient air environments but also play a crucial role under extremely harsh operating conditions such as high vacuum, extreme temperatures, intense radiation, high speeds, and high pressures [2,3].

Epoxy resin, characterized by its robust adhesion properties, minimal curing shrinkage, superior chemical stability, corrosion resistance, and excellent electrical insulation properties, represents a highly promising matrix material for polymer composite coatings [4]. The cured three-dimensional (3D) crosslinked network structure of epoxy resin confers it with outstanding mechanical properties and high chemical stability [5]. However, it also leads to drawbacks such as high brittleness, elevated wear rate, and an increased friction coefficient [6]. Consequently, the incorporation of suitable two-dimensional (2D) nanomaterials (such as graphene, transition metal dichalcogenides (TMDs), layered double hydroxides (LDHs), MXenes, etc.) into epoxy resin can effectively mitigate these inherent deficiencies, significantly enhancing its mechanical and tribological properties [7,8,9,10].

Molybdenum disulfide (MoS_2_), a member of the 2D TMDs family, exhibits a crystal structure akin to that of graphene. Each layer of MoS_2_ comprises a plane of Mo atoms sandwiched between two layers of S atoms. The weak van der Waals forces between adjacent layers lead to a well-structured stacking arrangement [11,12]. Therefore, MoS_2_ exhibits excellent lubrication performance and has been demonstrated as a promising solid lubricant additive. However, the application of MoS_2_ as a solid lubricant is limited by its poor oxidative stability, especially when used in humid air or at high temperatures, which limits its application as a solid lubricant. Enhancing its stability is thus a crucial objective to address this issue. JU et al. [13] used a radio frequency magnetron sputtering system to introduce nitrogen into MoS_2_ matrix, and the resulting Mo-S-N thin films significantly enhanced the oxidative resistance of MoS_2_ under high-temperature conditions, exhibiting superior mechanical and tribological properties compared to MoS_x_ films. Using a combination of MoS_2_ on the surface of in situ reduced graphene oxide (rGO), applied using a hydrothermal method, is also an acceptable approach to improve the stability of MoS_2_ coatings and also enhancing tribological performance [14]; the latter was attributed to the robust hybrid structure and synergistic effect between rGO and MoS_2_.

As a novel 2D material, hexagonal boron nitride (h-BN) shares certain common characteristics with graphene, such as excellent mechanical strength, toughness, good thermal conductivity, exceptional lubrication properties, and wear resistance, due to its similar structure [15,16,17]. Furthermore, h-BN also exhibits several unique advantages. For instance, its excellent electrical insulation can fundamentally prevent electrochemical corrosion, while its chemical stability enables it to demonstrate outstanding lubrication and antioxidation performance even under high-temperature conditions [18,19]. Bi et al. [20] prepared composite materials containing BN nanosheets (BNNSs) integrated with rGO via van der Waals forces without disrupting the structure of rGO. The ultra-thin BNNSs on the surface of rGO serves as a high-performance antioxidation coating in high-temperature oxidative atmospheres to prevent the oxidation of rGO.

In this work, we combined spherical MoS_2_ nanoparticles with BN nanosheets to fabricate a h-BN/MoS_2_ nanocomposite to serve as a nanolubricant for high-performance anti-wear epoxy coatings. Sheet-like MoS_2_ was converted into spherical MoS_2_ nanoparticles via a calcination method. Subsequently, the surface was modified by epoxy groups using the silane coupling agent 3-glycidoxypropyltrimethoxysilane (KH560, C_9_H_20_O_5_Si). Additionally, the h-BN was treated with a strong alkaline solution, and then the surface was modified using the silane coupling agent 3-aminopropyltetraethoxysilane (KH550, C_9_H_23_NO_3_Si), rendering its surface amino-functionalized. The “bridging effect” of the silane coupling agents KH550 and KH560, where the epoxy groups on KH560 react with the amino groups on KH550, established robust chemical bonds that uniformly decorated h-BN nanosheets onto the surface of MoS_2_. This modification and the introduction of h-BN not only enhanced the chemical stability of MoS_2_ but also improved the interfacial compatibility of the h-BN/MoS_2_ composite material in the epoxy resin, thereby enhancing the tribological performance of the composite coating. The effect of h-BN/MoS_2_ epoxy resin composite coatings on enhancing the tribological performance of the polymer matrix at various contents was investigated, and the frictional mechanism was revealed by analyzing the composition and microstructural evolution of the contact interface.

## 2. Experimental Section

### 2.1. Reagents and Materials

Epoxy resin E44 was purchased from Maclin Biochemical Technology Co., Ltd. (Shanghai, China). Thioacetamide was provided by Kelon Chemical Co., Ltd. (Chengdu, China). Sodium molybdate pentahydrate, acetone, and anhydrous ethanol were supplied by Aladdin Reagent Co., Ltd. (Shanghai, China). Silane coupling agents (KH560 and KH550) were obtained from Yuanye Bio-Technology Co., Ltd. (Shanghai, China). Concentrated hydrochloric acid was purchased from Lingfeng Chemical Reagent Co., Ltd. (Shanghai, China). N,N-Dimethylformamide was provided by Sinopharm Group Chemical Reagent Co., Ltd. (Shanghai, China). Methyl hexahydrophthalic anhydride was obtained from McLean (Hyderabad, India).

#### 2.1.1. Preparation of Spherical MoS_2_

Sodium molybdate dihydrate (Na_2_MoO_4_·2H_2_O, 5 mmol) and thioacetamide (CH_3_CSNH_2_, 30 mmol) were dissolved in deionized water (100 mL). The solution was stirred at room temperature for 30 min. The solution was then uniformly heated in a water bath at 90 °C for 5 min. After natural cooling to 85 °C, anhydrous ethanol (12 mL) was added while stirring, followed immediately by the slow addition of hydrochloric acid (20 mL). After cooling, the product in the dispersion was washed three times with deionized water and subsequently subjected to freeze-drying to yield MoS_X_ powder. The synthesized MoS_X_ was further processed by calcination in a tube furnace under an argon–hydrogen atmosphere. The temperature was increased at a heating rate of 2 °C/min, and after reaching 750 °C, the temperature was maintained for 50 min. After cooling down at a cooling rate of 5 °C/min, the spherical MoS_2_ powder was obtained [21].

#### 2.1.2. Preparation of KH560-MoS_2_

The spherical MoS_2_ powder (0.2 g) was dispersed in anhydrous ethanol (80 mL) and sonicated for 30 min. Simultaneously, the silane coupling agent KH560 (0.2 g) was gradually added dropwise to deionized water (80 mL) while stirring for 30 min. The dispersed KH560 solution was then added dropwise into the MoS_2_ dispersion and transferred to a three-necked flask. The mixture was heated at 80 °C in a water bath for 3 h. Afterward, it was washed sequentially with ethanol and deionized water by centrifugation three times each. The precipitate was freeze-dried and ground into powder. This process yielded the surface-modified spherical MoS_2_, denoted as KH560-MoS_2_.

#### 2.1.3. Preparation of KH550-BN

Hexagonal boron nitride (h-BN, 0.5 g) was dispersed in a 5 M NaOH solution (50 mL) and stirred at room temperature for 30 min. The dispersion was then transferred to a 100 mL PTFE reactor and subjected to a hydrothermal reaction at 120 °C for 12 h. Following this, the product was washed, isolated, and freeze-dried to obtain hydroxylated boron nitride (BN-OH). The obtained BN-OH powder and the silane coupling agent KH550 were used as raw materials, to prepare the functionalized BN (KH550-BN) by the same means as KH560-MoS_2_.

#### 2.1.4. Preparation of h-BN/MoS_2_ Hybrid

The KH560-MoS_2_ (0.4 g) was dispersed in 35 mL N,N-Dimethylformamide (DMF) solvent and subjected to sonication for 30 min to achieve uniform dispersion. Simultaneously, the KH550-BN (0.1 g) was added in DMF solvent (35 mL), to form the uniform suspension. The uniformly dispersed KH560-MoS_2_ and KH550-BN suspension were mixed together and then 180 mL of DMF was added again. After stirring for 30 min, the mixture was transferred to a three-necked flask and subjected to heating in an oil bath at 105 °C for 5 h [22,23]. After cooling to room temperature, the resulting mixture was separated by centrifugation. The centrifugation parameters utilized in this procedure were as follows: a rotational speed of 8000 rpm maintained for a period of 10 min. The obtained precipitate of h-BN/MoS_2_ composite was washed three times with ethanol to remove residual solvent. Finally, the h-BN/MoS_2_ composite material was dispersed in ethanol to form a dispersion solution for further use.

### 2.2. Preparation of h-BN/MoS_2_ Epoxy Resin Composite Coating

The 30 mm × 30 mm × 1 mm 304 stainless steel plates were selected as substrates for coating. All substrates were initially pretreated with a 200-grit and 600-grit sandpaper sequentially to ensure sufficient surface roughness, then, followed by ultrasonic cleaning with ethanol. The h-BN/MoS_2_ mixture was ultrasonically treated in acetone for 30 min to achieve a relatively uniform dispersion. Subsequently, suspensions of h-BN/MoS_2_ with mass fractions of 1%, 2%, 5%, and 10% were individually added to 2 g epoxy resin E44. After ultrasonic dispersion for 30 min, the curing agent (1 mL) was introduced to each mixture. The mixtures were then stirred for 5 h to ensure thorough dispersion. Finally, the resulting mixtures were applied to the pretreated surfaces of the 304 stainless steel plates and maintained at 35 °C for 30 min. The curing process involved sequential steps at 80 °C for 120 min and 120 °C for 240 min, yielding epoxy resin composite coatings with varying contents of h-BN/MoS_2_.

### 2.3. Characterization

The crystalline structures of powder samples were analyzed using an X-ray diffraction (XRD, D8 Advance, Bruker, Billerica, MA, USA) device with Cu Kα radiation (λ = 1.5406 Å). The crystalline phases in the diffractograms were indexed using the International Centre for Diffraction Data (ICDD) database, specifically the PDF-4+ 2023 release. Transmission electron microscopy (TEM) and high-resolution transmission electron microscopy (HRTEM, FEI Tecnai F30, Thermo Fisher Scientific, Hillsboro, OR, USA, acceleration voltage: 300 kV) with an energy dispersive X-ray spectroscopic (EDS) detector are used to observe the detailed microstructure. X-ray photoelectron spectroscopy (XPS, PHI QUANTERA II (ULV AC JAPAN Ltd., Chigasaki, Kanagawa, Japan)) with monochromatic Al Kα radiation (150 W, 500 µm, and 1486.6 eV) was employed to monitor the chemical composition. Scanning electron microscopy (SEM, JSM-7001 F, (JEOL Ltd., Tokyo, Japan)) was used to observe the surface morphology and microstructure of the chemical composition, and the morphology and composition of the wear scar surfaces were analyzed with an energy-dispersive spectrum (EDS). Raman spectroscopy (inVia, Renishaw plc, Wotton-under-Edge, Gloucestershire, UK), utilizing a 532 nm excitation source, was employed to identify the structural and chemical composition of the samples. The hardness of the coatings was characterized using a microVickers hardness tester (MHVD-1000IS, Shanghai Jujing Precision Instrument Manufacturing Co., Ltd., Shanghai, China). In the test, the sample was placed with the surface of the coating with the wear scar on top and fixed in the sample stage. The hardness of the wear scar surface can be obtained by converting the indentation depth of the indenter on the wear scar surface. In order to ensure the accuracy of the test results, three different coatings were selected, and each coating sample was measured 5 times at different locations. Fourier-transform infrared spectroscopy (FTIR, Nicolet IS10, Thermo Fisher Scientific, Waltham, MA, USA) was used to characterize the surface functional groups of the test samples, with the wavenumber ranging from 400 cm^−1^ to 4000 cm^−1^.

### 2.4. Tribological Properties Tests

The tribological properties of the composite coatings were evaluated using a vertical universal friction and wear testing machine (MMW-1, manufactured by Jinan Outuo Testing Equipment Co., Ltd., Jinan, China). The experiment employed a rotational friction method, with a friction radius of 5 mm, a friction linear velocity of 0.2 m/s, a load of 30 N, and a test duration of 30 min. High-carbon chromium bearing steel balls (GCr15) with a diameter of 5 mm were used as the counter material. To ensure the reliability of the results, the experiments were repeated at least three times under the same conditions, and the average values of friction coefficient and wear rate were calculated. The formula for the specific wear rate is as follows:*W* = 2*πRS*/*NVT*(1)
where *W* (mm^3^ N^−1^ m^−1^) represents the wear rate of the sample, *R* is the rotational radius (mm), *V* is the linear speed (m/s), *N* is the load (N), and *S* is the cross-sectional area of the wear scar (mm^3^).

## 3. Results and Discussion

### 3.1. Characterization of Materials

The preparation process of h-BN/MoS_2_ is illustrated in Figure 1. Initially, spherical MoS_2_ nanoparticles were prepared from MoS_2_ nanosheets by calcination, followed by surface modification with KH560. The molecular structure of KH560 contains a siloxane bond (Si-O) and an organic functional group (epoxy), wherein the siloxane bond forms stable covalent bonds with the MoS_2_ surface via hydrolysis and condensation reactions (Figure 1a). Subsequently, h-BN nanosheets are treated with a strong alkaline solution to generate active sites such as hydroxyl groups (-OH) on their surface, facilitating subsequent chemical reactions. The molecular structure of KH550 contains amino groups (-NH_2_) and ethoxy groups (-OEt); the ethoxy groups undergo hydrolysis in the presence of water to form hydroxyl groups, which then condense with the active sites on the h-BN surface to form stable -Si-O-BN bonds, resulting in KH550-modified h-BN, termed KH550-BN (Figure 1b). Finally, through the “bridging effect”, h-BN is decorated on the MoS_2_ surface. The crosslinking reaction between KH560 and KH550 molecules facilitates the combination of these two distinct nanomaterials, ultimately yielding a composite material with h-BN uniformly distributed on MoS_2_ (Figure 1c).

The XRD patterns of the BN, KH550-BN, MoS_2_, KH560-MoS_2_, and h-BN/MoS_2_ composites are shown in Figure 2a. The diffraction peaks of h-BN were indexed to the hexagonal phase (PDF# 00-034-0421). For BN and KH550-BN, characteristic peaks can be observed at 2θ values of 26.72°, 41.67°, and 43.95°, corresponding to the (002), (100), and (101) crystal planes of h-BN, respectively [24]. The reflections of MoS_2_ were matched to the standard card (PDF# 00-037-1492). In the XRD patterns of MoS_2_ and KH560-MoS_2_, diffraction peaks appear at 2θ = 14.2°, 33.1°, 39.4°, and 58.7°, corresponding to the (002), (100), (103), and (110) crystal planes of MoS_2_ in the 2H phase structure, respectively [25]. It is evident that the peak positions of BN and MoS_2_ nanoparticles modified by the silane coupling agent remain unchanged from the original position, with only a slight decrease in peak intensity. The modification process does not alter the crystal structure of the nanoparticles, only their surface properties. The XRD pattern of h-BN/MoS_2_ displays characteristic peaks for both MoS_2_ and BN, confirming the successful synthesis of h-BN/MoS_2_ nanocomposite materials.

Infrared spectroscopy (IR) analysis confirms the successful grafting and reaction of various functional groups during different treatment steps. The Figure 2b displays the IR spectra of pure BN, BN-OH, and KH550-BN. Pure BN displays two distinct absorption peaks at 1315 cm^−1^ and 766 cm^−1^, corresponding to the in-plane B-N stretching vibration and the out-of-plane B-N-B bending vibration, respectively [26]. In addition, BN-OH hydroxylated by NaOH exhibits an -OH stretching vibration peak at 3440 cm^−1^, indicating that the hydroxyl groups are successfully grafted on the surface of BN [27]. The ultrasonic treatment of BN-OH grafted with a silane coupling agent (KH550-BN) results in a blue shift in the infrared characteristic peak positions, with a slightly weaker peak intensity compared to the original BN. This indicates that the BN layers in KH550-BN are effectively exfoliated, leading to reduced interlayer interactions and consequently an increase in vibration frequency. The FTIR spectrum of KH550-BN reveals several key peaks: a Si-O stretching vibration peak at 934 cm^−1^, asymmetric vibrations of Si-O-Si and Si-O-C at 1004 cm^−1^ and 1087 cm^−1^, and asymmetric and symmetric stretching vibrations of -CH_2_- and -CH_3_ at 2870 cm^−1^ and 2894 cm^−1^, respectively. A broad peak near 3300 cm^−1^ may indicate -NH_2_ groups at the edges of KH550-BN, and an absorption peak at 1625 cm^−1^ corresponds to the characteristic -NH- stretch. These absorption peaks confirm the successful grafting of the silane coupling agent KH550 onto the surface of BN nanoparticles [28,29]. The IR spectra of pure MoS_2_, KH560-MoS_2_, and h-BN/MoS_2_ composites are shown in the Figure 2c. Due to the poor absorbance of MoS_2_, the peak intensities are not prominent, with characteristic peak concentrated around 400 cm^−1^. For KH560-MoS_2_, the asymmetric vibrations of Si-O-Si and Si-O-C can be seen at 1043 cm^−1^ and 1082 cm^−1^, and the asymmetric and symmetric stretching vibrations of -CH_2_- and -CH_3_ can be seen at 2978 cm^−1^ and 2905 cm^−1^, respectively. These absorption peaks confirm the successful grafting of the silane coupling agent KH560 on the surface of MoS_2_ nanoparticles [30,31,32]. For h-BN/MoS_2_ composites, the disappearance of the amino peaks from KH550-BN and the epoxy peaks in KH560-MoS_2_ indicates effective grafting through functional group reactions.

By analyzing the Raman spectra of MoS_2_ before and after KH560 treatment (Figure 2d), additional structural information about the MoS_2_ grafts can be obtained. Untreated MoS_2_ exhibits two main peaks at 379.34 cm^−1^ and 403.83 cm^−1^, corresponding to the in-plane E^1^_2g_ and out-of-plane A^1^_g_ vibration models of MoS_2_ structure, respectively [33]. After KH560 treatment, the E^1^_2g_ and A^1^_g_ vibrational models of KH560-MoS_2_ shift to higher frequencies, with the E^1^_2g_ showing a greater shift than the A^1^_g_. This indicates that KH560 molecules impose tensile strain on the MoS_2_ layer, affecting the expansion and vibration. Furthermore, this observation confirms that the silane coupling agent KH560 is successfully grafted and modified on the surface of MoS_2_.

The chemical states of elements in KH550-BN and KH560-MoS_2_ were characterized by X-ray photoelectron spectroscopy (XPS). Figure 3a presents the full XPS spectrum of the h-BN/MoS_2_ composite, where the C1s, O1s, and Si2p peaks are prominently observed. These peaks are attributed to the presence of KH560 and KH550, indicating that two silane coupling agents successfully participate in the formation of the composite material. The C1s peak of the h-BN/MoS_2_ composite can be deconvoluted into four Gaussian curves (Figure 3d), with the peaks at 284.4, 285, 285.9, and 288 eV corresponding to C-C, C-N, C-O, and C=O bonds, respectively [34]. The possible bonding reactions between the amino groups in KH550 and the epoxy groups in KH560 may form C-N and C=O bonds, indicating chemical interactions. Furthermore, in the composite material h-BN/MoS_2_, the N1s peak of C-N is located at 399.05 eV (Figure 3e), showing a significant shift compared to that in KH550-BN. In addition, in the composite material h-BN/MoS_2_, the Mo3d (Figure 3c) peaks of Mo3d_3/2_, Mo3d_5/2_, and S2s (Figure 3f) are located at 232, 228.8, and 226.1 eV, respectively, exhibiting noticeable shifts compared to KH560-MoS_2_ [35]. This suggests changes in the chemical environments of N and Mo, confirming the occurrence of chemical reactions between KH550 and KH560. The results demonstrate that a successful composite formation of the BN and MoS_2_ composite material (h-BN/MoS_2_) can be achieved through the chemical reaction between KH550 and KH560.

The scanning electron microscopy (SEM) images of h-BN/MoS_2_ composites are shown in the Appendix A. The composites are in the form of irregular spheres with a diameter of about 68.99 ± 14.87 nm. The transmission electron microscopy (TEM) (Figure 4a) overview reveals that ultrathin BN is successfully coated on MoS_2_, with differences in lateral dimensions forming a uniform structure of BN-modified spherical MoS_2_. Meanwhile, the use of high-resolution scanning transmission electron microscopy (HRTEM) allows us to observe different regions with distinct boundaries, where the dark region refers to MoS_2_ and the bright region refers to BN, indicating the formation of interfaces in h-BN/MoS_2_.

The magnified HRTEM image (Figure 4d) shows that the 0.63 nm stripe spacing can correspond to the (002) crystal plane of MoS_2_ in phase I (Figure 4b), and the 0.166 nm crystal plane spacing obtained through Fourier transform can match the (004) crystal plane of BN in phase II (Figure 4c), which can be further confirmed by the selected area electron diffraction (SAED) diagram (Figure 4e). The Figure 4f shows the energy dispersive X-ray spectroscopy (EDS) elemental map analysis of h-BN/MoS_2_. The results of EDS show that the constituent elements B and N are uniformly distributed throughout the selected area, indicating that BN is uniformly distributed on the surface of MoS_2_.

### 3.2. Mechanical and Tribological Properties

As illustrated in Figure 5, the inclusion of h-BN/MoS_2_ composites notably decreases the coefficient of friction (COF) of the epoxy composite coating. As shown in Figure 5a,b, with the increase in h-BN/MoS_2_ composite content, the COF first decreases and then increases. When the h-BN/MoS_2_ content is 5 wt %, COF reaches the minimum of 0.026. Compared with the pure EP coating, the COF decreases by 88%. The COF is reduced by 75% compared to a single MoS_2_ coating. For pure BN, due to the obvious agglomeration of 5 wt % BN in the composite coating, the COF increases sharply, and the friction process is in a state of instability. At the same time, it can be found that with an increase in h-BN/MoS_2_ content, the wear-resistant and friction-reducing performance of the epoxy composite coating improves further. However, when the filler content is excessive, the agglomeration of h-BN/MoS_2_ nanoparticles will occur, affecting its dispersion in the epoxy resin. Subsequently, the agglomeration will cause stress and friction-related heat concentration, which produces defects in the epoxy resin matrix. Under the influence of shear stress during the friction process, the epoxy composite coating becomes more susceptible to peeling off, leading to an increase in the size of wear debris entering the friction interface. Consequently, this results in an elevated COF [36].

The addition of h-BN/ MoS_2_ significantly reduces the COF of the epoxy composite coating. This is because MoS_2_ and h-BN have excellent self-lubricating properties. However, in the friction process, the local high temperature causes the self-lubrication property of the single MoS_2_ coating to fail, while for the composite h-BN/MoS_2_, h-BN is uniformly coated on the surface of MoS_2_. The high chemical stability and strong oxidation resistance of h-BN make MoS_2_ resistant to local high temperature oxidation, and the coating therefore has a lower COF and better wear resistance. Because MoS_2_ is a spherical nanoparticle assembled in the form of flakes, the sliding friction changes to a rolling friction to a certain extent, which reduces the friction coefficient of the friction interface. Furthermore, spherical MoS_2_ particles crushed by shear force become flakes again, filling the cracks at the friction interface and repairing the interface damage, which improves the anti-friction performance of the composite coating. At the same time, the h-BN/MoS_2_ composite material falls off from the epoxy composite coating and enters the friction interface in the friction process, which improves the lubrication effect of the friction interface, thus improving the friction-reducing performance of the composite coating [37].

As shown in Figure 6a, the addition of nano-fillers reduces the specific wear rate and width of the abrasion mark in the epoxy composite coating compared to pure EP coatings, thus enhancing wear resistance. As the h-BN/MoS_2_ content of the composite material increases, the wear rate and abrasion mark width initially decrease but then increase. At a 5 wt % h-BN/MoS_2_ composite material content, the wear rate measures 1.5 × 10^−6^ mm^3^ N^−1^ m^−1^, which is 99% lower than that of the pure EP coating (1.8 × 10^−4^ mm^3^ N^−1^ m^−1^). At the same time, the wear width is only 637 μm, which is 70.3% lower than that of pure EP coating (2149 μm). This reduction is attributed to the high specific surface area of the h-BN/MoS_2_ composite material, which promotes a strong interface interaction with the epoxy resin and enhances the load-bearing capacity of the epoxy composite coating. Additionally, the self-lubricating properties of MoS_2_ and BN minimize wear on the composite coating. The uniform distribution of hard BN on MoS_2_ further reduces the wear rate and width of the composite coating. However, as the h-BN/MoS_2_ content of the composite increases, nanoparticle agglomeration decreases the load-bearing capacity of the composite coating, leading to an increase in wear rate and width. Nonetheless, these values remain lower than those of pure EP coatings.

As illustrated in Figure 6b, the variation in the COF of h-BN/MoS_2_ composite material fillers with different contents corresponds to differences in hardness values. The hardness of the 5 wt % h-BN/MoS_2_ composite coating measures 36.84 HV. Among the h-BN/MoS_2_ composite coatings, the one with the highest hardness value exhibits a 52% increase in hardness compared to the pure EP coating. This indicates that the h-BN/MoS_2_ composite material significantly enhances the stiffness of the epoxy resin composite coating. This improvement is attributed to the increased crosslinking density of the epoxy resin due to the nano-fillers. As the hardness of the epoxy resin composite coating increases, its plastic deformation capacity decreases, leading to enhanced wear resistance. However, an excess of nano-fillers can cause particle agglomeration, thereby reducing the interface adhesion between the filler and the epoxy resin. This can introduce defects, elevate stress and heat concentration, diminish the optimization performance of crosslinking density, reduce the stiffness of the composite coating, weaken its load-bearing capacity, and enhance wear-through susceptibility, consequently generating more wear debris.

As illustrated in Figure 6c, the long-cycle tribological performance test of the 5 wt % h-BN/MoS_2_ epoxy resin composite coating shows that the COF remains stable at approximately 0.05 for the first 10,000 s. During this period, it can be inferred that a friction film forms between the steel ball of the upper friction pair and the resin film on the composite coating surface. This transition changes the friction mechanism from direct contact between the steel ball and the composite coating to sliding between the friction film and the resin film.

The TGA and DTG curves of pure EP, 5 wt % MoS_2_/EP, and 5 wt % h-BN/MoS_2_/EP under N_2_ and air atmospheres are presented in the Figure 7. It can be observed that the initial decomposition temperatures of the 5 wt % MoS_2_/EP and 5 wt % h-BN/MoS_2_/EP are 385 °C and 393 °C, respectively, which are higher than that of pure EP (367 °C) under N_2_ atmospheres. This indicates that the addition of h-BN/MoS_2_ can effectively improve the thermal stability of EP. It can also be seen that the DTG curves (Figure 7c) of the three materials show very similar trends, with each material only showing one clear thermal decomposition peak. This situation suggests that the thermal decomposition processes of these three materials have similar mechanisms. The thermal decomposition temperatures of the 5 wt % MoS_2_/EP and the 5 wt % h-BN/MoS_2_/EP are 422.8 °C and 427.6 °C, respectively, which are significantly higher than the thermal decomposition temperature of pure EP (420 °C) under N_2_ atmospheres. The 5 wt % h-BN/MoS_2_/EP also exhibits excellent thermal stability under air atmosphere. This demonstrates that the addition of h-BN/MoS_2_ can significantly enhance the heat resistance of EP, while the addition of h-BN effectively improves the thermal conductivity of EP, thereby enhancing the thermal stability of EP. Improving the thermal stability of EP can effectively prevent thermal damage to the material during the friction process, thereby optimizing its tribological performance to some extent.

The morphology of the fracture surface of the coating was observed using SEM (Figure 8). From Figure 8a, the surface of the pure EP coating appears smooth, and some furrows of typical brittle fractures are visible, indicating that the coating is brittle and the fracture toughness is very low [38]. Compared with the EP coating, the composite coating exhibits obvious plastic deformation, which enhances its toughness. The composite coating containing only MoS_2_ (Figure 8b) shows a regular orientation of crack extension, with the inter-crack area appearing very smooth. This indicates rapid crack propagation, which is not conducive to impeding crack growth. With an increase in h-BN/MoS_2_ content, cracks in the composite coating propagate along the layers, resulting in a rougher fracture surface with enhanced plastic deformation and protrusions. At a h-BN/MoS_2_ content of 5% (Figure 8e), the fracture surface of the composite coating becomes more intricate and multi-directional, presenting a smoother and larger fractured area that effectively inhibits crack propagation. This improvement can be attributed to the improved dispersion of h-BN/MoS_2_ and the outstanding mechanical properties of h-BN, which enhance the toughness and plasticity of the coating [39,40].

The SEM images and wear surface analyses are presented in Figure 9. For the pure EP coating, the wear surface details align with the optical microscope images, showing numerous large wear debris fragments and spalling pits. These features contribute to the brittle behavior and poor tribological performance of the epoxy coating. The primary mechanisms of wear that can be observed are typical fatigue wear and adhesive wear. With the addition of a single MoS_2_ material (Figure 9e), compared with the pure EP coating (Figure 9f), only a small amount of debris is observed on the wear surface of the composite coating, and the wear surface becomes relatively flat and produces a dense and smooth MoS_2_ friction film [41]. However, due to the poor mechanical strength of MoS_2_, the MoS_2_ friction film is not continuous, and some microcracks can still be observed due to the fatigue wear of the epoxy resin.

The incorporation of h-BN/MoS_2_ in the composite results in a notably even and unblemished wear surface, demonstrating superior wear resistance. Augmenting the proportion of the composite material markedly ameliorates the friction and wear characteristics of the coating, bolsters the load-bearing capability of the EP matrix, and fosters the creation of an uninterrupted h-BN/MoS_2_ surface lubrication layer on the sliding surface, thereby mitigating the production of substantial wear debris [42]. Nevertheless, an excessive presence of the composite material could instigate the agglomeration of h-BN/MoS_2_ nanoparticles, which may amplify fatigue wear and adhesive wear during frictional processes, culminating in a significant quantity of wear debris and peeling pits on the wear surface of the composite coating. This could consequently impinge upon its tribological performance.

The wear surface of the 5 wt % h-BN/MoS_2_ composite coating (Figure 9c) is almost without debris and microcracks, and the wear surface is uniformly covered by a continuous friction film, indicating that the h-BN/MoS_2_ composite coating has excellent tribological properties. The main reason for the excellent tribological properties of the 5 wt % h-BN/MoS_2_ composite coating is that h-BN/MoS_2_ is uniformly dispersed in the epoxy resin matrix, which makes up for the pore defects of the resin matrix. At the same time, the boron nitride nanosheets are uniformly coated on—and have a good synergistic effect with—the surface of molybdenum disulfide, which enhances the chemical stability of the molybdenum disulfide, maintains its self-lubrication effect, and improves the performance of the MoS_2_ composite coating at high sliding speeds and heavy loads. In addition, the “bridging effect” of the coupling agent helps to improve the interface compatibility between h-BN/MoS_2_ composite material and EP matrix, further promoting the stress transfer from EP to the h-BN/MoS_2_ composite material.

To further investigate the wear mechanism of h-BN/MoS_2_ epoxy resin composite coatings, XPS analysis was performed on the 5 wt % h-BN/MoS_2_ epoxy resin composite coating after the frictional test. As shown in Figure 10, the C1s spectrum (Figure 10a) reveals four peaks at 284.7 eV, 285.3 eV, 286.6 eV, and 288.9 eV, corresponding to C-C, C-O, C-N, and C=O bonds in the h-BN/MoS_2_ composite material or epoxy resin matrix. For the Mo3d signal (Figure 10b), peaks at 229.2 eV and 232.8 eV are attributed to Mo^4+^3d_5/2_ and Mo^4+^3d_3/2_ in MoS_2_, respectively. The peaks at 231.6 eV and 234.7 eV correspond to Mo^6+^3d_5/2_ and Mo^6+^3d_3/2_ in MoO_3_, indicating partial oxidation of MoS_2_ during the friction process, which leads to the formation of MoO_3_ [43]. The S2p spectrum (Figure 10c) shows that the peak at 168.8 eV is attributed to the S^6+^ in FeSO_4_, while the peaks at 164.1 eV and 162.8 eV come from FeS_2_ and MoS_2_, respectively. Furthermore, the binding energy of Fe2p (Figure 10e) consists of six peaks, which belong to Fe_3_O_4_ (724 eV), FeSO_4_ (714 eV), FeS_2_ (712 eV), and Fe_2_O_3_ (709 eV). These results indicate that a solid and self-lubricating transfer film composed of MoS_2_ and h-BN was formed during the friction process of the steel ball and the composite coating, and the tribo-chemical reaction products are MoO_3_, FeS_2_, Fe_2_O_3_, FeSO_4_, and Fe_3_O_4_ metal compounds [44,45,46].

Based on the above results and analysis, the schematic diagram of enhancement mechanisms of h-BN/MoS_2_ on the tribological properties of the EP composite coating are shown in Figure 11. During friction, the steel ball in the friction pair adheres to a small amount of epoxy resin and simultaneously peels off part of the resin, generating debris. This cyclic process of resin adhesion and debris peeling results in exacerbated adhesive wear. Furthermore, fatigue wear of the resin leads to the propagation of microcracks on the coating surface, culminating in the formation of numerous pits and debris on the friction surface. On the one hand, with the help of the coupling agent’s “bridging effect”, the h-BN/MoS_2_ epoxy resin composite coating enhances the interfacial compatibility between the h-BN/MoS_2_ composite material and epoxy resin, making them uniformly dispersed in the epoxy resin matrix and hindering the aggregation of nano-inorganic materials. On the other hand, the uniform modification of h-BN improves the chemical stability of MoS_2_. It effectively protects molybdenum disulfide from oxidation, reduces the generation of molybdenum trioxide during friction, and maintains its self-lubricating properties. Under repeated stress, due to the synergistic effect between h-BN and MoS_2_, the h-BN with high mechanical properties and MoS_2_ with self-lubrication easily form a uniform and tough transfer film on the friction pair surface, transforming the friction sliding between the friction pair and the coating into mutual sliding between the transfer films. Therefore, the h-BN/MoS_2_ epoxy resin composite coating exhibits excellent anti-friction and wear-resistant properties.

## 4. Conclusions

In summary, a novel h-BN/MoS_2_ composite material was successfully synthesized by calcining spherical MoS_2_ nanoparticles generated by flaky self-assembly and utilizing the “bridging effect” of a silane coupling agent to uniformly and vertically modify h-BN nanosheets onto the surface of spherical MoS_2_. The composite material was introduced into the EP matrix to prepare a EP-h-BN/MoS_2_ solid lubricating coating. Compared with the pure EP coating and the composite coating reinforced by a single h-BN or MoS_2_, the h-BN/MoS_2_ composite coating had relatively excellent tribological performance. When the content of the h-BN/MoS_2_ composite material was 5 wt %, the composite coating had a lower average friction coefficient and specific wear rate, at 0.026 and 1.5 × 10^−6^ mm^3^ N^−1^ m^−1^. This is mainly due to the high mechanical performance and reinforcement stability of h-BN, which successfully protects MoS_2_ during friction, avoids its oxidation, and hinders friction to maintain the durability of its self-lubrication effect. At the same time, there is a good synergistic effect between h-BN and MoS_2_, which makes the wear surface and the upper friction pair form a uniform and continuous transfer film during friction, enhancing the friction and wear performance of the composite coating. Therefore, the prepared h-BN/MoS_2_ composite material has broad application prospects in the field of polymer-based self-lubricating composite coatings.

## Figures and Tables

**Figure 1 nanomaterials-15-00401-f001:**
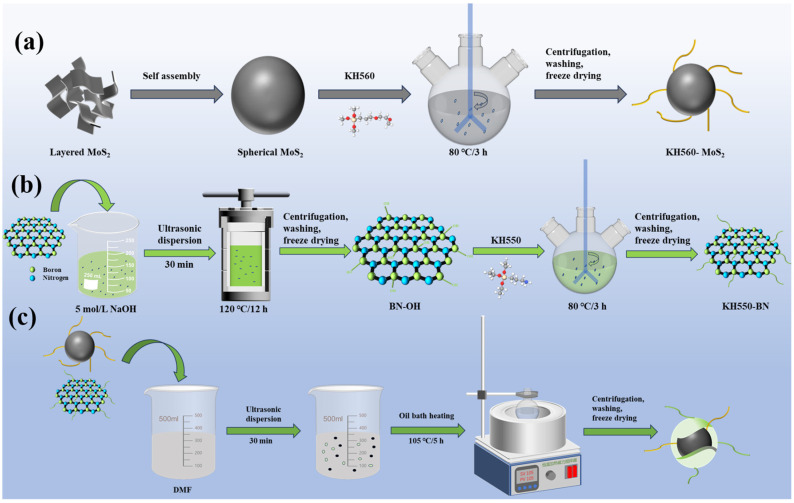
The schematic diagrams of the preparation processes for (**a**) KH560-MoS_2_, (**b**) KH550-BN, and (**c**) h-BN/MoS_2_.

**Figure 2 nanomaterials-15-00401-f002:**
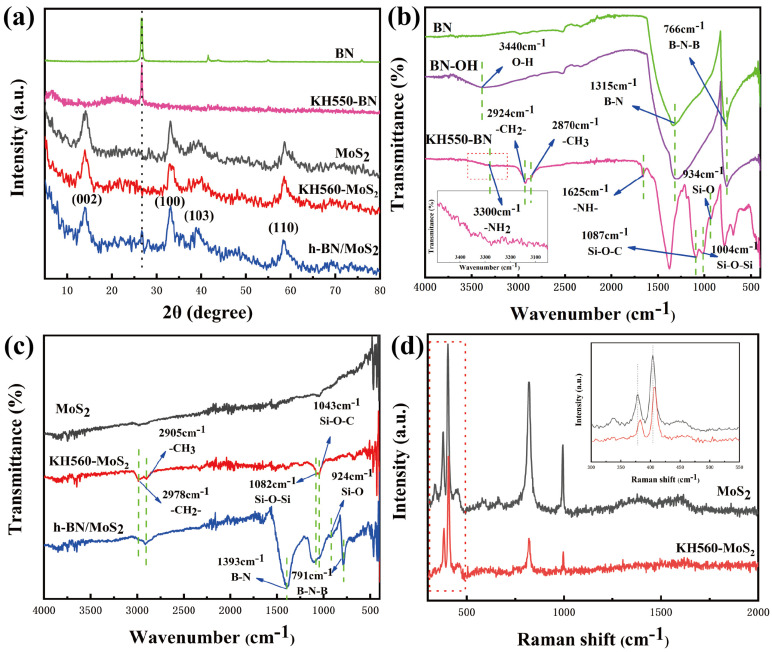
(**a**) XRD patterns of BN, KH550-BN, MoS_2_, KH560-MoS_2_, and h-BN/MoS_2_ hybrids, FT-IR spectra of (**b**) BN, BN-OH, KH550-BN, and (**c**) MoS_2_, KH560-MoS_2_, and h-BN/MoS_2_ hybrids, and (**d**) Raman spectroscopy of MoS_2_ and KH560-MoS_2_.

**Figure 3 nanomaterials-15-00401-f003:**
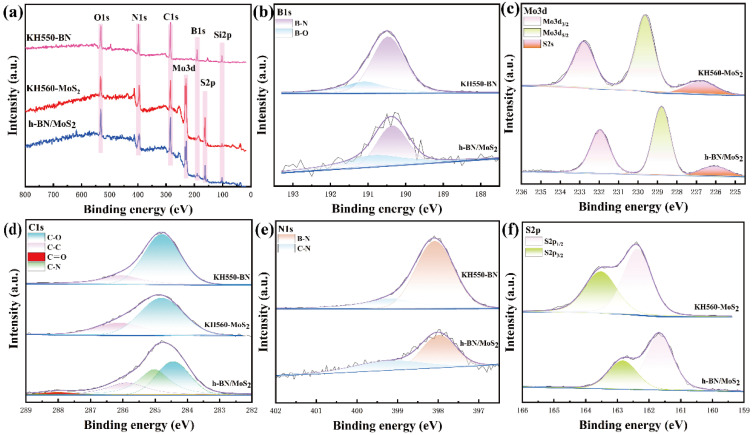
(**a**) Full-range XPS spectra and high-resolution spectra of h-BN/MoS_2_ hybrids including (**b**) B 1s, (**c**) Mo 3d, (**d**) C 1s, (**e**) N 1s, and (**f**) S 2p.

**Figure 4 nanomaterials-15-00401-f004:**
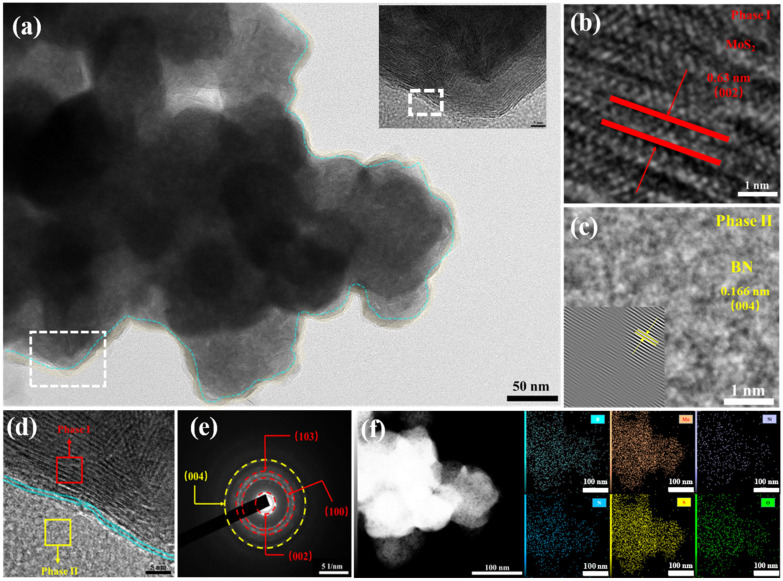
(**a**) TEM images of h-BN/MoS_2_ hybrids, (**b**–**e**) HRTEM image, and the corresponding SAED patterns of h-BN/MoS_2_ hybrids. (**f**) Elemental mapping of h-BN/MoS_2_ hybrids.

**Figure 5 nanomaterials-15-00401-f005:**
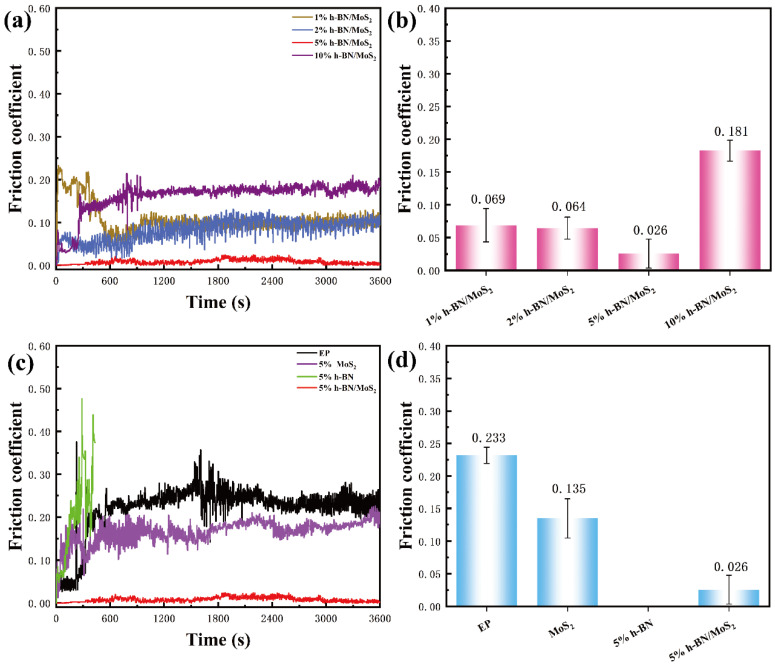
(**a**,**c**) The friction coefficients of the samples as a function of sliding time. (**b**,**d**) The friction coefficients’ error bar for different samples.

**Figure 6 nanomaterials-15-00401-f006:**
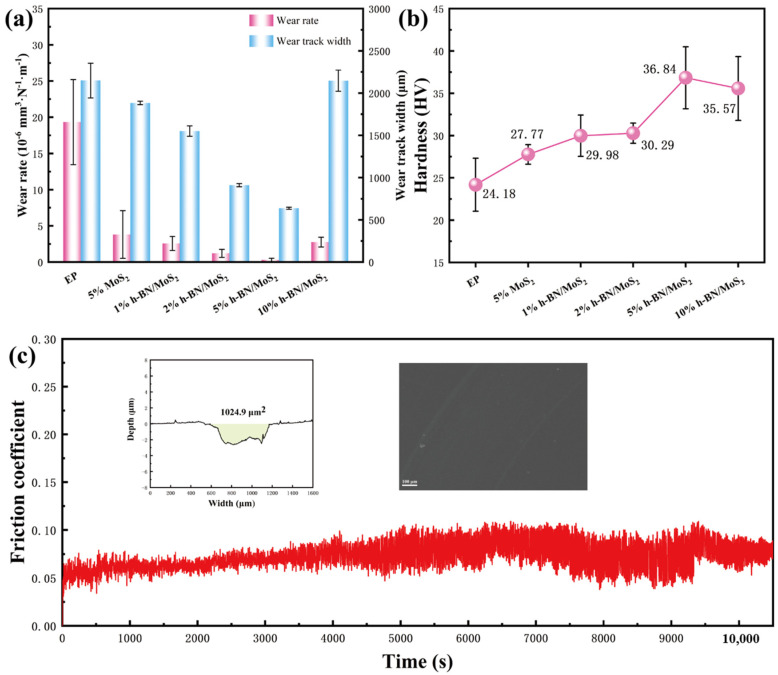
(**a**) Wear rates and wear track width of the coatings for different samples. (**b**) Vickers hardnesses of the coatings with different samples. (**c**) The long-cycle tribological performance of 5 wt % h-BN/MoS_2_.

**Figure 7 nanomaterials-15-00401-f007:**
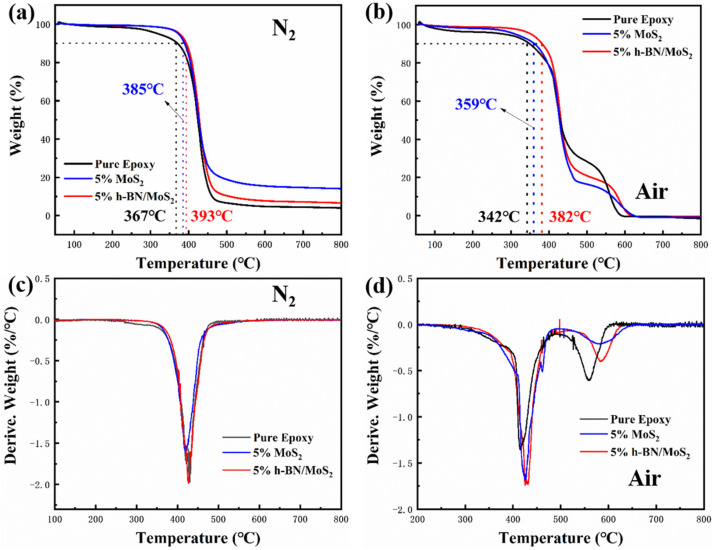
(**a**,**b**) TGA and (**c**,**d**) DTG curves of pure EP, 5 wt % MoS_2_/EP, and 5 wt % h-BN/MoS_2_ /EP under N_2_ and air atmospheres.

**Figure 8 nanomaterials-15-00401-f008:**
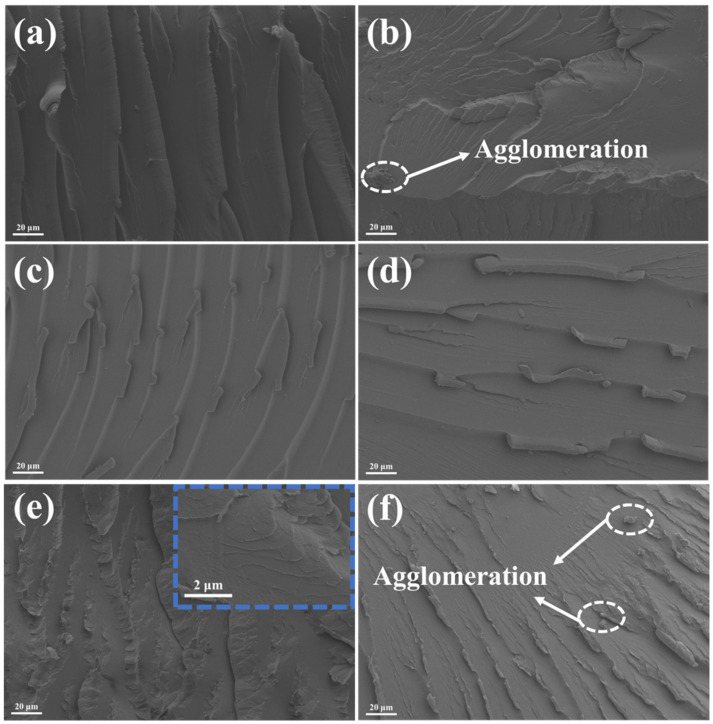
Fracture sections for different coatings: (**a**) EP, (**b**) 5 wt % MoS_2_/EP, (**c**) 1 wt % h-BN/MoS_2_/EP, (**d**) 2 wt % h-BN/MoS_2_/EP, (**e**) 5 wt % h-BN/MoS_2_/EP, and (**f**) 10 wt % h-BN/MoS_2_/EP.

**Figure 9 nanomaterials-15-00401-f009:**
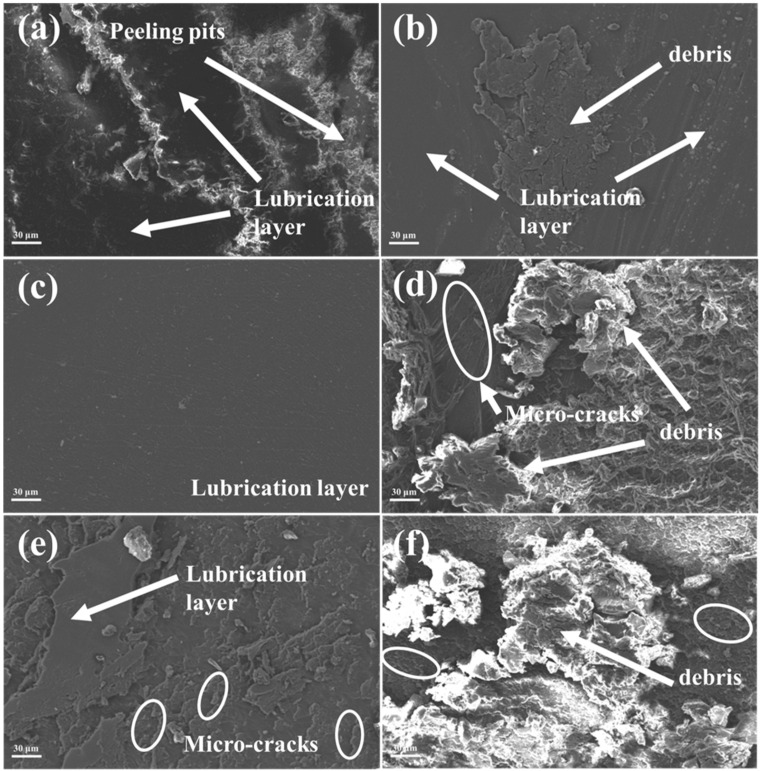
SEM morphology of worn surfaces of the composites with different contents of h-BN/MoS_2_ (**a**) 1 wt %, (**b**) 2 wt %, (**c**) 5 wt %, (**d**) 10 wt %), (**e**) the pure EP coating, and (**f**) the single MoS_2_ coating.

**Figure 10 nanomaterials-15-00401-f010:**
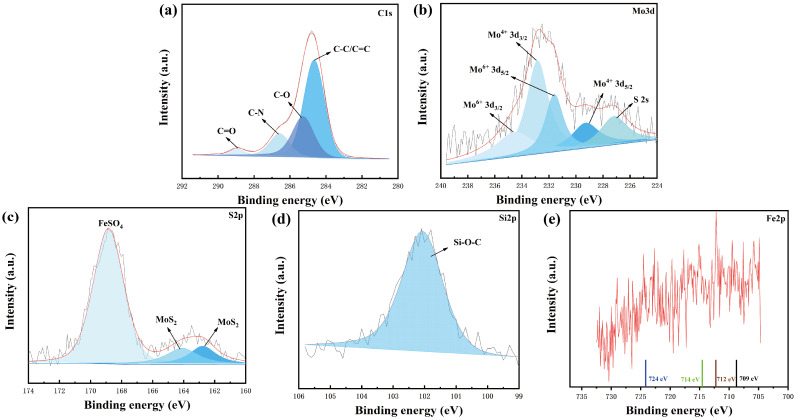
XPS spectra of 5 wt % h-BN/MoS_2_ composite coating after friction: (**a**) C 1s, (**b**) Mo 3d, (**c**) S 2p, (**d**) Si 2p, and (**e**) Fe 2p.

**Figure 11 nanomaterials-15-00401-f011:**
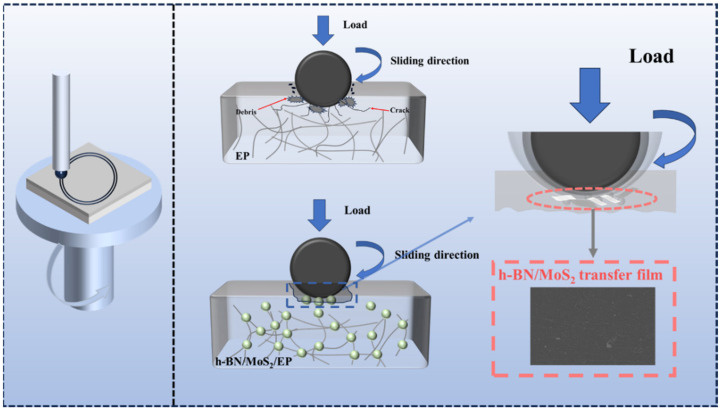
Schematic of wear mechanism of EP composite coatings enhanced via h-BN/MoS_2_ hybrids.

## Data Availability

The original contributions presented in this study are included in the article/Appendix A. Further inquiries can be directed to the corresponding author(s).

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
