# Peer review of "Tribological Behavior and Mechanism of Silane-Bridged h-BN/MoS2 Hybrid Filling Epoxy Solid Lubricant Coatings"

_nanomaterials, 2025, doi:10.3390/nano15050401_

Round 1

Reviewer 1 Report

Comments and Suggestions for Authors

Is it possible to ducument the statement that MoS2 after the sliding test transforms to flakes and fills the wear groves and reapirs the sliding surface.

Check the figure 6 label, instead od (d) it should be (c)

Also figure 6b is not related to COF at all.

Please consider changing the paper structure. Several times in paper some statement were explained and by images supported much later. Maybe you cuold try to separate results and discussion on following parts: structural, mechanical and tribological.

Reviewer 2 Report

Comments and Suggestions for Authors

The objective of the manuscript “Tribological Behavior and Mechanism of Silane-Bridged h-BN/MoSâ‚‚ Hybrid Filling Epoxy Solid Lubricant Coatings” was to synthesize and characterize a novel h-BN/MoSâ‚‚ composite through the lamellar self-assembly of spherical MoSâ‚‚ particles and the uniform and vertically oriented deposition of h-BN, using a silane coupling agent. Additionally, the study systematically investigated the chemical composition and microstructure of the composite, as well as its influence on the tribological properties of epoxy coatings, focusing on reducing the friction coefficient and specific wear rate. The goal was to enhance the wear resistance and chemical stability of the material for friction applications. The manuscript is well-written and well-organized, and I recommend it for publication with minor corrections.

In the following sentence “After cooling, the solution was washed three times with deionized water and subsequently subjected to freeze-drying to yield MoSX powder”, how is a solution washed?

The following sentence “Afterward, it was washed multiple times with ethanol and deionized water, followed by centrifugation” needs to be rewritten. “Multiple times” does not help with the reproducibility of the work. Additionally, what was the sequence of washing between ethanol and deionized water?

In the following sentence “After cooling to room temperature, the resulting mixture was separated by centrifugation”, what were the centrifugation conditions?

In the sentence “The obtained precipitate of h-BN/MoSâ‚‚ composite was washed multiple times with ethanol to remove residual solvent”, again, resolve the “multiple times” issue, as it makes the work irreproducible.

In the following sentence “The 30 mm×30 mm 304 stainless steel plates are selected as substrates for coating”, the word “are” should be replaced with “was”. Review the verb tense throughout the manuscript. Additionally, a unit is missing in the dimensions of the stainless steel plates.

In the following sentence “The h-BN/MoSâ‚‚ mixture is ultrasonically treated in acetone for 30 minutes to achieve a relatively uniform dispersion”, correct the verb tense!

In Section 2.4, specify the database used for indexing the crystalline phases in the diffractograms.

The correct way to describe the equipment used in the research is: model, brand, city, and country of manufacture. Apply this format to all equipment mentioned in the manuscript.

For the hardness and wear tests, mention details such as load, time, and the type of indenter used.

In the sentence “The tribological properties of the composite coating are evaluated by the vertical universal friction and wear testing machine (MMW-1, manufactured by Jinan Outuo Testing Equipment Co., Ltd.)”, correct the verb tense.

Again, correct the verb tense in the following sentence “Initially, spherical MoSâ‚‚ nanoparticles are prepared from MoSâ‚‚ nanosheets by calcination, followed by surface modification with KH560.”

In the discussion of the diffractograms, include the crystallographic card numbers used to index the crystalline phases.

Comments on the Quality of English Language

The English grammar needs to be improved.

Reviewer 3 Report

Comments and Suggestions for Authors

The study is solid and the manuscript is well organized and written.

I found several issues which should be addressed to improve the manuscript quality. These comments are listed below:

1)      Please explain the MoS2 calcination conditions. It is not clear how 780 ºC could be reached in 50 min at 2ºC/min rate.

2)      The first paragraph in Section 3 and Figure 1 can be better moved to Section 2 since they describe the method and not the results.

3)      Line 374. Is their any evidence for the formation of the transfer layer? If not, the author should state that this is only a hypothesis.

4)      It is difficult to conclude from Fig. 9 that the structures are cracks.

5)      The results suggesting tribochemical reaction on the contact surfaces are convincing. I wonder if some gaseous products could be formed in such reactions and analysed to be able directly observe the tribochemical reaction pathways. I acknowledge the authors comments with respect.

6)      It would be interesting to show XPS data for the resin with only MoS2 additive to compare the protection effect of hBN against MoS2 oxidation. This would support the authors argument on the oxidation protection of the decoration hBN layer.

7)      Please check the marks in the manuscript showing typo and language errors.

Round 2

Reviewer 3 Report

Comments and Suggestions for Authors

The authors have made efforts to improve the manuscript addressing all questions and comments. I support the publication of the manuscript. I also would like to draw author's attention to the following recently published work which addresses the effect of MoS2 on properties of polymer-matrix composites used as solid lubricants: https://www.sciencedirect.com/science/article/pii/S0301679X25001069